# OpenReview forum: "Uncertainty-Aware Decision Transformer for Stochastic Driving Environments"
_robot-learning.org/CoRL/2024/Conference — CoRL 2024_

### Official Review · Reviewer_Jtpp · 2024-07-06
**good paper**

**Originality:** 3
**Technical Quality:** 3
**Clarity Of Presentation:** 3
**Potential Impact:** 4
**Recommendation:** 3
**Confidence:** 4

**Review:**

### Strengths:
- **Innovative Approach**: Introducing uncertainty estimation directly into decision transformers is a novel approach that addresses a critical challenge in RL.
- **Practical Relevance**: The focus on autonomous driving scenarios enhances the practical relevance and potential impact of the research.
- **Experimental Rigor**: Comprehensive experiments across diverse driving scenarios validate the effectiveness and robustness of UNREST.
- **Clear Presentation**: The paper is generally well-structured and clearly presents its methodology and findings, facilitating understanding and replication.

### Weaknesses:
- **Complexity of Uncertainty Estimation**: The mathematical formulation of uncertainty estimation using conditional mutual information could be challenging for readers not familiar with information theory.
- **Limited Comparison**: While UNREST demonstrates superior performance, comparison with a wider range of state-of-the-art methods or alternative transformer adaptations could further strengthen the paper.
- **Generalization**: The extent to which UNREST can generalize to other domains beyond driving scenarios remains an open question.

**Quality Of The Limitations Section:**

1

**Questions For Rebuttal:**

I'm not an expert for  self-driving research. I try to give some suggestions:

1. There are many methods improving Decision Transformer, e.g., TIT [1], PDiT [2], STEER [3] and their applications for Traffic Signal Control [4,5]. I think it will be better to compare these more powerful methods rather than just comparing with DT and TT.

2. In many general RL setting, the environment is also stochastic, could the proposed UNREST be used in Traffic Signal Control or other RL settings? This could improve the potential impact on other fields.

[1] Transformer in Transformer as Backbone for Deep Reinforcement Learning. Arxiv 2022.
[2] PDiT: Interleaving Perception and Decision-making Transformers for Deep Rinforcement Learning. AAMAS 2024.
[3] Sequential Asynchronous Action Coordination in Multi-Agent Systems: A Stackelberg Decision Transformer Approach. ICML 2024.
[4] CoSLight: Co-optimizing Collaborator Selection and Decision-making to Enhance Traffic Signal Control. KDD 2024.
[5] X-Light: Cross-City Traffic Signal Control Using Transformer on Transformer as Meta Multi-Agent Reinforcement Learner. IJCAI 2024.

**Robotics Focus:**

2

**Summary Of Paper:**

The paper introduces UNREST, a novel approach for offline reinforcement learning (RL) in uncertain driving scenarios. Traditional RL models struggle in stochastic environments where identical actions may lead to varied outcomes. UNREST addresses this by integrating uncertainty estimation directly into decision-making. It measures uncertainty through conditional mutual information between transitions and returns, emphasizing actual outcomes over assumed transitions. This approach mitigates 'uncertainty accumulation' and 'temporal locality' issues typical in driving environments. Notably, UNREST replaces global returns with truncated versions less sensitive to environmental variability, enhancing robustness. Experimental results across diverse driving scenarios demonstrate UNREST's superior performance and highlight the efficacy of its uncertainty-aware planning strategy.

**Summary Of Recommendation:**

The recommendation for weak accept is grounded in several key factors. Firstly, the paper exhibits high quality evidenced by its innovative approach integrating uncertainty estimation into decision transformers, addressing a significant challenge in RL. The clarity of presentation, while generally effective, could benefit from clearer explanations in mathematical formulations. Originality shines through in adapting transformer models for stochastic environments without additional complex models, enhancing its significance in autonomous driving. The experimental rigor and practical relevance underscore its potential impact. Addressing complexities in uncertainty estimation and broadening comparative analyses would further strengthen its contribution to the field of autonomous systems and reinforcement learning.

---

### Official Review · Reviewer_1GUa · 2024-07-21
**Uncertainty estimation to disentangle effect of stochastic environment transitions on reward**

**Originality:** 4
**Technical Quality:** 5
**Clarity Of Presentation:** 5
**Potential Impact:** 3
**Recommendation:** 4
**Confidence:** 4

**Review:**

The paper is well written with clear explanations and concise notation to follow through. My questions are posted in the 'Questions For Rebuttal' section.

Strengths
Explanations are clear without reliance on extensive jargon and equations. The Reinforcement learning formulation is standard.
The positioning of work with respect to past related work is commendable. Past work was explained clearly and how the authors' method differs from those was also succinctly described
The ablation experiments are extensive and demonstrate a clear advantage of the various modifications over the baseline DT method.
The figures are easy to follow (except Fig 4) and do a good job at summarizing the method.

Weaknesses
The appendix is too long.
The limitations are not addressed in the main paper, although it's present in the appendix.
I was unable to follow the example in Section 5.4 Uncertainty Visualization as the figure was unclear due to lack of legends and marginally inappropriate choice of colors.

**Quality Of The Limitations Section:**

3

**Questions For Rebuttal:**

1. The uncertainty estimation measures the discrepancy in estimating the reward from information before the transition (absence of s_t) and information of the transition (presence of s_t), is my understanding correct? perhaps, the notation in equation 3 was a bit confusing.
2. Please move the limitations section in the main paper instead of being in the appendix.
3. I didn't understand how the 'lightweight uncertainty prediction model' in section 4.4 is trained, perhaps more details would be great.
4. How is the uncertainty threshold epsilon selected?
5. How is the hyperparameter c (for frequent switching between certain and uncertain segments) selected?
6. Perhaps, section 5.4 Uncertainty Visualization could be expanded to show more instances of uncertainty on ego vehicle behavior.
7. How is the hyperparameter n (upper percentile of target return for planning) selected?

**Robotics Focus:**

3

**Summary Of Paper:**

The main idea of the paper is to avoid generative modelling or adversarial training for handling uncertainty in rewards due to stochastic transitions. This is done via estimating the effect of a transition on the truncated return via conditional mutual information between transitions and returns.

**Summary Of Recommendation:**

It's a well written paper with very minor revisions needed for rebuttal.

---

### Official Review · Reviewer_hnBQ · 2024-07-26
**Review of paper 220**

**Originality:** 3
**Technical Quality:** 3
**Clarity Of Presentation:** 3
**Potential Impact:** 2
**Recommendation:** 3
**Confidence:** 4

**Review:**

The paper is well written and in combination with the supplementary material is presented with a lot of interesting experiments and ablation studies.  Although there are no real world robot results, the simulation results presented seem convincing compared to the baselines presented.

My first concern is about the dataset collected. The dataset uses an inbuilt CARLA autopilot to collect the expert trajectories - what steps were taken to make sure that there was enough environment stochasticity in the dataset? From what I understand the stochasticity here is the uncertainty over the ado agent behaviors (and not different towns or weather conditions). For example, in the turning example presented in figure 1, the uncertainty in making the right turn without collision comes from different behaviors that the ado agent exhibits (adversarial vs cooperative). Did the authors explicitly model different ado agent behaviors? was there any stochasticity added to the autopilot behavior?

During inference, to estimate the uncertainty a KD-tree is used to mimic the uncertainty output from the return transformers. Presenting some details and evaluating the performance of the KD-tree like accuracy could be useful to understand the sensitivity of the planner to errors in uncertainty estimation.

**Quality Of The Limitations Section:**

3

**Questions For Rebuttal:**

1) One of the return transformers - p_phi_s (T_ret, s_t) - depends on the truncated return (T_ret) and the state at timestep t (s_t), but the truncated return also has the current state s_t as an input, what is the reason for having s_t twice?
A broader question is, why cant the return transformer just be a probabilistic Q-function (under the expert policy) and then use the variance captured by it to identify certain vs uncertain states?

2) How sensitive is UNREST to novel driving scenarios? i.e., scenarios that were not present in the dataset to learn the uncertainty estimators. For example, lets say there is a novel roundabout environment that wasn't present in the dataset. Is the uncertainty estimate less reliable? In uncertain regions, UNREST 'rests' on expert actions, but what if we don't have expert actions?

3)  UNREST is evaluated on a few stochastic D4RL environments to demonstrate generalizability, what are examples of temporal locality in these environments/tasks analogous to driving scenarios?

**Robotics Focus:**

3

**Summary Of Paper:**

The paper aims to address a key challenge with decision transformer (DT), which is sensitive to stochasticity in the environment leading to poor performance.  The two key observations that authors use are 1) Uncertainty accumulation and 2) Temporal locality of policies, they suggest the use of truncated returns instead of global returns to account for overly optimistic actions from DT's in stochastic environments. The truncation is done using an estimate of uncertainty. Extensive evaluation is done in CARLA simulation to validate the results.

**Summary Of Recommendation:**

Weak accept

---

### Author Rebuttal · Authors · 2024-08-07

**We submit the revised pdf here, left responses to each reviewer in respective official reviews.**

**Note that if the network connection is poor, Markdown formulas in the comments may not display correctly and might show as raw characters instead. If this occurs, switching to a more stable network and refreshing the page should resolve the issue.**

---

### Decision · Program_Chairs · 2024-09-04

**Decision:**

Accept

**Comment:**

This paper introduces uncertainty into a decision transformer, to better equip the model to handle stochastic RL environments. Reviewers are all generally positive about this work.

Strengths:
* Well written, simple but good improvement over base DT, method seems like a good idea - a nice extension

Weaknesses:
* Only tested in driving scenarios (CARLA) - additional detail needed around the test conditions here and evidence for stochasticity in environment, unclear whether this would generalise.
* Some clarity issues in figures, requirements to refer to appendix for detail, limitations not discussed in main paper.

For the rebuttal, I recommend answering the reviewers technical questions, and addressing concerns around experimental tests and potential generality.

Post rebuttal:
The authors did a good job of addressing remaining concerns through their rebuttal and answering questions around test conditions and baselines. I think this paper has merit and is a valuable contribution.